

# Density matrices in quantum gravity

**Tarek Anous**[1★]**, Jorrit Kruthoff**[2†] **and Raghu Mahajan**[3‡]

**1** Institute for Theoretical Physics and $\Delta$-Institute for Theoretical Physics,
University of Amsterdam, Science Park 904, 1098 XH Amsterdam, The Netherlands
**2** Department of Physics, Stanford University, Stanford, CA 94305-4060, USA
**3** Institute for Advanced Study, Princeton, NJ 08540, USA

★ t.m.anous@uva.nl, † kruthoff@stanford.edu, ‡ raghum@ias.edu

## Abstract

We study density matrices in quantum gravity, focusing on topology change. We argue that the inclusion of bra-ket wormholes in the gravity path integral is not a free choice, but is dictated by the specification of a global state in the multi-universe Hilbert space. Specifically, the Hartle-Hawking (HH) state does not contain bra-ket wormholes. It has recently been pointed out that bra-ket wormholes are needed to avoid potential bags-of-gold and strong subadditivity paradoxes, suggesting a problem with the HH state. Nevertheless, in regimes with a single large connected universe, approximate bra-ket wormholes can emerge by tracing over the unobserved universes. More drastic possibilities are that the HH state is non-perturbatively gauge equivalent to a state with bra-ket wormholes, or that the third-quantized Hilbert space is one-dimensional. Along the way we draw some helpful lessons from the well-known relation between worldline gravity and Klein-Gordon theory. In particular, the commutativity of boundary-creating operators, which is necessary for constructing the alpha states and having a dual ensemble interpretation, is subtle. For instance, in the worldline gravity example, the Klein-Gordon field operators do not commute at timelike separation.

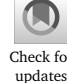

# 1 Introduction

The gravitational path integral requires not only a sum over metrics on a manifold of a given topology, but also a sum over manifolds of different topologies.[1] One strong reason to include a sum over topologies in the gravitational path integral is that doing so gives us the Hawking-Page transition in Anti-de Sitter space [2], which, famously, is the bulk dual of the confinement-deconfinement transition in gauge theory [3,4].[2]

While the topological classification of three- and higher-dimensional manifolds is quite complicated, in two-dimensional gravity theories, such as worldsheet string theory [7] or Jackiw-Teitelboim (JT) gravity [8–10], the sum over topologies reduces to a sum over the Euler characteristic of the manifold, a single integer. Recently, there has been a surge of interest in the sum over different topologies in the gravitational path integral. For instance, in [11,12] it was argued that spacetime wormholes are crucial in understanding entropy paradoxes in black hole physics. The sum over different topologies also plays a crucial role in the interpretation of JT gravity as a random matrix theory, and the computation of the ramp and plateau regions of the spectral form factor [13–15].

A systematic way to quantize a theory that involves topology change and multiple universes is the so-called "third quantization" formalism [16]. Third quantization is a bad name for reasons that will become clear later in this note, but we stick with it because of legacy. The name string field theory (which is the field theory of multiple strings) suggests that "universe field theory" would perhaps be a more appropriate name.[3] This formalism has been reviewed and clarified in a recent paper by Marolf and Maxfield [17]. An incomplete list of references is [18–24].

Given the recent interest in this subject, one of our aims in this brief note is to clarify some of the conceptual features of third quantization. Much of the mystery of this formalism disappears if we keep in mind the example of worldline gravity, where the third-quantized theory is just the Klein-Gordon theory, as we review in section 2. The worldline gravity discussion also shows us that the commutativity of boundary-creating operators, which is necessary for constructing the alpha states and having a dual ensemble interpretation, is subtle. For example, the Klein-Gordon field operators do not commute at timelike separation. However, as usual, if the target space (which, in general, would be "superspace") is analytically continued to Euclidean signature, we do expect the boundary-creating operators to commute.[4] We also point out that the general construction of the baby universe Hilbert space is the same as the Gelfand-Naimark-Segal (GNS) construction in algebraic QFT.

In order to talk about quantities like the entropy of the density matrix of the universe, we first need to specify a global state in the multi-universe Hilbert space. There are multiple options for the global state, as we review in section 3. One feature of different global density matrices, defined by the gravitational path integral, is whether or not manifolds connecting the bra and the ket, or a bra-ket wormholes, exist. The choice of whether or not to include bra-ket wormholes is not a free choice, rather it depends on which global state we pick. Two special global states are the Hartle-Hawking state [25,26], which does not have bra-ket wormholes, and the Page state [27], which includes bra-ket wormholes. Most of the observations in these sections 2 and 3 are not new and have already appeared in the early literature on quantum cosmology; our aim is to present them in a way that we found useful.

---

[1]For an opposing viewpoint, arguing not to include topology change, see for example [1].

[2] However, note that there is still room for a sum over only certain topologies, as long as the sum is unambiguously defined. For example, in two dimensions, we can choose to sum either over just the oriented manifolds, or to sum over oriented and unoriented manifolds. The recent papers [5,6] featured a sum over a restricted class of three-manifolds, the so-called handlebodies.

[3] We thank Juan Maldacena for suggesting this name.

[4] Superspace has infinitely many negative-signature directions and this analytic continuation might be subtle.

Section 4 is devoted to some implications of these observations. Perhaps most interestingly, we find an application for the notion of effective wormholes, due to [28]. The bra-ket wormholes that are needed to resolve the bags-of-gold type paradoxes in de Sitter (as in section 6 of [12]) and strong subadditivity paradoxes [29] would need to "emerge" in the Hartle-Hawking state, since they are not originally present in the definition of the Hartle-Hawking state. One way this can happen is if the density matrix of the Hartle-Hawking state, restricted to a single, late-time classical universe, contains effective wormholes obtained by tracing out the unobserved universes. More extreme possibilities are that the Hartle-Hawking state could be non-perturbatively gauge equivalent to a state with bra-ket wormholes, or that the baby universe Hilbert space is one-dimensional. In section 4.2 we discuss how boundary entropy computations in AdS are conceptually different, and emphasize that the replica wormholes in that case are simply the wormholes between the various ket boundaries in the bulk Hartle-Hawking state.

Throughout this paper, by the baby universe Hilbert space, or the third-quantized Hilbert space, we will mean the Hilbert space that was denoted $\mathcal{H}_{\text{BU}}$ or $\mathcal{H}_{0,0}$ in [17].

*Note added:* While this article was nearing completion, the paper [30] appeared which has some overlap with our section 2.2.

## 2 Third quantization through the lens of worldline gravity

Let us recall some basic facts regarding 1d gravity. We specialize to a class of theories describing worldlines embedded in Minkowski space $\mathbb{R}^{1,d-1}$ [7,16]. The worldline fields are an einbein $e(\tau)$ and $d$ scalar fields $x^\mu(\tau)$ with action

$$I_{\text{WL}}[e, x^\mu] = \int_{\tau_i}^{\tau_f} \mathrm{d}\tau \left( e^{-1} \dot{x}^\mu \dot{x}^\nu \eta_{\mu\nu} - me \right). \tag{1}$$

This theory is invariant under reparametrizations of the worldline time $\tau$. Apart from its simplicity, the main advantage of this theory is that the "third-quantized" theory describing multiple 1d universes [16] is just the Klein-Gordon scalar field theory in $d$-dimensional Minkowski space.[5] If the worldlines are not allowed to split, we get a free theory. By allowing a worldline to branch into two, and two worldlines to fuse into one, we can get a $\phi^3$ theory in target space.

A typical computation in this theory would be to sum over all worldlines, or 1d manifolds $X$, with boundary $\partial X$ consisting of $n+m$ points, and each boundary point labelled by a point in $\mathbb{R}^{1,d-1}$. Picking points $x_1, \ldots, x_m \in \mathbb{R}^{1,d-1}$ to be the "past" boundaries, and $y_1, \ldots, y_n \in \mathbb{R}^{1,d-1}$ to be the "future" boundaries, we denote this quantity by

$$\langle y_1 \ldots y_n \, | \, x_1 \ldots x_m \rangle := \int_{X:|\partial X|=n+m} e^{-I_{\text{WL}}} = \quad . \tag{2}$$

Note that these quantities are the precise analogs of the quantities $\langle Z[\tilde{J}_1] \ldots Z[\tilde{J}_n] \, | \, Z[J_1] \ldots Z[J_m] \rangle$ considered in [17] (see their equation (2.5)).[6] These analogies and others that will be discussed below are summarised in table 1.

---

[5]Usually, the Klein-Gordon theory would be obtained as a "second quantization" of single-particle wavefunctions, without any reparametrization invariance involved at any stage. Indeed, the worldline perspective and

Table 1: Comparison between objects in worldline gravity, whose third quantization leads to Klein-Gordon theory, and higher-dimensional gravity theories whose third-quantized description should perhaps be called universe field theory, imitating the name string field theory. The word superspace is used like it is used in quantum cosmology, it is the space of spatial metrics modulo spatial diffeomorphisms. In general, $\mathfrak{S}$ is the space of boundary conditions in the "worldvolume path integral", and each element of $\mathfrak{S}$ represents a boundary condition that can be imposed on a single connected component of the boundary. The notation involving $Z[J]$'s is due to [17].

| Worldline gravity/Klein-Gordon theory | Worldvolume gravity/Universe field theory |
|---|---|
| $\mathfrak{S} = \mathbb{R}^{1,d-1}$ | $\mathfrak{S} = \text{superspace}$ |
| $x \in \mathbb{R}^{1,d-1}$ | $J \in \text{superspace}$ |
| $\lvert \text{KG-vac} \rangle$ | $\lvert \text{HH} \rangle$ |
| $\lvert x \rangle = \phi(x) \lvert \text{KG-vac} \rangle$ | $\lvert J \rangle = \phi(J) \lvert \text{HH} \rangle$ or $\lvert Z[J] \rangle = \widehat{Z[J]} \lvert \text{HH} \rangle$ |
| $\langle y \lvert x \rangle$ | $\langle J_1 \lvert J_2 \rangle$ or $\langle Z[J_1] \lvert Z[J_2] \rangle$ |
| $\langle \text{KG-vac} \lvert \phi(x) \phi(y) \lvert \text{KG-vac} \rangle$ | $\langle \text{HH} \lvert \phi(J_1) \phi(J_2) \lvert \text{HH} \rangle$ or $\langle \text{HH} \lvert \widehat{Z[J_1]} \, \widehat{Z[J_2]} \lvert \text{HH} \rangle$ |
| $-\Box + m^2 = 0$ | $H_{\text{WdW}} = 0$ |

The simplest nonzero quantity of the form (2) corresponds to just having one point in the past and one point in the future. It is well-known that this computes the Klein-Gordon propagator:[7]

$$\langle y \lvert x \rangle = \langle \text{KG-vac} \lvert \phi(y) \phi(x) \lvert \text{KG-vac} \rangle, \tag{3}$$

with $\lvert \text{KG-vac} \rangle$ the standard Klein-Gordon vacuum. Similarly, the general quantity in (2) will compute an $(n+m)$-point function in the Klein-Gordon theory. The relation (3) and similar expressions for the higher point functions make it clear that the usual Klein-Gordon vacuum state $\lvert \text{KG-vac} \rangle$ is the precise analog of the Hartle-Hawking state $\lvert \text{HH} \rangle$ [25], and also that the field operators $\phi(x)$ are the precise analogs of what were called the $\widehat{Z[J]}$ operators in [17]. See table 1 and section 3 for some more details. With the benefit of hindsight and because of this direct analogy to the Klein-Gordon theory, perhaps $\phi(J)$ is a better notation for $\widehat{Z[J]}$. Below we will use $\phi(J)$ and $\widehat{Z[J]}$ interchangeably.

Null states, which played a crucial role in [17], also exist in this 1d gravity model. To see this, note that the field operator $\phi(x)$ is labeled by a point $x$ in $\mathbb{R}^{1,d-1}$, it is not restricted to a single Cauchy slice. However, the hyperbolic nature of the Klein-Gordon equation allows us to relate the field operator $\phi(t, \mathbf{x})$ to a linear combination of $\phi(0, \mathbf{x})$, and these relationships give rise to null states. The Klein-Gordon equation in this model is the analog of the Wheeler-DeWitt equation in higher-dimensions.

In general, if we have some reparametrization-invariant path integral with some set $\mathfrak{S}$ of allowed boundary conditions, then we should consider $\mathfrak{S}$ as our "target spacetime" and the

---

the Schwinger proper-time representation of the propagator is useful even in practical QFT calculations [31] or exploring the analyticity structure of Feynman diagrams (see Chapter 18 of [32]).

[6] In analogy to the well established notation in worldline gravity, we would prefer to denote these quantities as $\langle \tilde{J}_1 \ldots \tilde{J}_n \lvert J_1 \ldots J_m \rangle$, without the letter $Z$. The reason in [17] for adopting the notation $Z$ was the anticipation of a boundary dual, but from the perspective of the bulk path integral, it is perhaps more natural to omit the letter $Z$.

[7] Depending on the range of integration of the lapse variable, we can either get two-point functions that obey the homogeneous Klein-Gordon equation, or two-point functions that obey the Klein-Gordon equation with a delta-function source. For details, see, for example [33]. Notice also that the states $\lvert x \rangle$ are not orthonormal. See also [34] for an analogous computation of the off-shell propagator for strings in a special case.

collection of field operators would be $\phi(J)$ for every $J \in \mathfrak{S}$. The role of $\phi(J)$ is to insert a boundary in the "worldvolume" path integral with boundary condition $J$. Note also that in general the space $\mathfrak{S}$ is infinite dimensional, has no symmetries, and can have an infinite number of negative signs in its metric signature.

For $D$-dimensional gravity,[8] the space $\mathfrak{S}$ is called superspace and consists of all $(D-1)$-metrics $h_{ij}(\sigma)$ modulo parallel diffeomorphisms. Here $\sigma$ denotes the collection of $D-1$ coordinates on the worldvolume. The Wheeler-DeWitt equation is a functional differential equation, and it resembles a Laplacian on a space with an infinite number of timelike directions. It is a well-known fact that the conformal factor of $h_{ij}(\sigma)$ for each $\sigma$ corresponds to a time-like direction in $\mathfrak{S}$, along which the line-element is negative. One can further speculate whether there is some notion analogous to a Cauchy slice in QFT, i.e. a subset $\mathfrak{C} \subset \mathfrak{S}$ such that the collection of field operators $\phi(J)$ with $J \in \mathfrak{C}$ constitutes the linearly independent field operators.

To make things a bit more concrete, a useful example to keep in mind is JT gravity with a positive cosmological constant. This theory was discussed in detail in [35, 36] (see also [37]) and describes the fluctuations of a boundary mode in rigid $dS_2$. The superspace $\mathfrak{S}$ consists of a variable corresponding to the boundary length $\ell$ (which is all that remains of $h_{ij}(\sigma)$ after gauge-fixing the spatial diffeomorphisms), and the dilaton profile $\Phi(\sigma)$. In the minisuperspace approximation, where $\Phi$ is taken to be constant, $\mathfrak{S}$ reduces to two-dimensional Minksowki space and the Wheeler-DeWitt equation is a Klein-Gordon equation for a massive charged scalar in an electric field. The null states would therefore be analogous to the ones found in the 1d gravity theory.

## 2.1 Commutativity of operators, and operator products

A very important claim of [17] is that, for any $J_1, J_2 \in \mathfrak{S}$, the operators $\widehat{Z[J_1]}$ and $\widehat{Z[J_2]}$ commute. These operators can thus be simultaneously diagonalized to yield a basis of the so-called alpha states. In the boundary dual description, each alpha parameter labels a member of an ensemble of theories, and the eigenvalue of $\widehat{Z[J]}$ in an individual alpha state is interpreted as the numerical value of the partition function $Z[J]$ in a specific boundary theory.

The argument in [17] (see the discussion around their equation (2.16)) for the commutativity of $\widehat{Z[J_1]}$ and $\widehat{Z[J_2]}$ for any $J_1, J_2 \in \mathfrak{S}$ operators is based on the fact that exchanging the two boundaries corresponding to $J_1$ and $J_2$ does nothing to the path integral. We want to point out that this argument is perhaps too quick, and that the statement of commutativity deserves a more careful analysis.

Recall that in the Klein-Gordon theory, we have

$$[\phi(t_1, \mathbf{x}_1), \phi(t_2, \mathbf{x}_2)] \neq 0 \tag{4}$$

if the points $(t_1, \mathbf{x}_1)$ and $(t_2, \mathbf{x}_2)$ are timelike separated. This is a seeming counterexample to the above claim about the commutativity of any two of the $\widehat{Z[J]}$ operators. Note however that if the target space in the Klein-Gordon theory is Wick-rotated to Euclidean signature, these operators do commute. This fact will be discussed more completely in an upcoming publication [38], where it will be argued that, at least in this respect, the universe field theory is more similar to QFT in Euclidean signature.[9] Here, we just want to point out that this is a subtle issue that involves analytic continuations in the target space $\mathfrak{S}$. In this context, see also section 8 of [24] (especially the discussion around their equation (8.5)) which takes the view that the boundary-creating operators do not commute in general, even in higher-dimensional gravity.

---

[8]Note that we are deliberately using $D$ here, rather than $d$, to distinguish it from the target space of worldline gravity.

[9]We thank Don Marolf and Henry Maxfield for correspondence on this point.

It is interesting to note that the theories dual to AdS are conformal, and hence their partition functions only depend on the conformal class of the boundary metric, with the anomaly coefficients of the boundary encoded in the bulk coupling constants. In other words, the dependence of $Z[J]$ on the "timelike" directions in $\mathfrak{S}$ seems to be completely determined from a subset $\mathfrak{C} \subset \mathfrak{S}$. Thus, in order to retain the ensemble interpretation needed in AdS, it might be enough that the field operators $\widehat{Z[J_1]}$ and $\widehat{Z[J_2]}$ commute only whenever $J_1, J_2 \in \mathfrak{C} \subset \mathfrak{S}$. This last property is indeed true even in the Lorentzian Klein-Gordon theory with $\mathfrak{C} = \mathbb{R}^{d-1}$ and $\mathfrak{S} = \mathbb{R}^{1,d-1}$.

Another interesting question is about singularities in operator products. It is well-known that operator products in Euclidean QFT have singularities as the two operator insertions becomes coincident. These singularities become branch cuts on the lightcone when continued to Lorentzian signature, and these branch cuts give rise to nonzero commutators when operators become timelike separated.

Relatedly, if we are given a list of correlators $\langle \text{KG-vac} | \phi(t_1, \mathbf{x}_1) \dots \phi(t_n, \mathbf{x}_n) | \text{KG-vac} \rangle$ for all $n$ and all $(t_i, \mathbf{x}_i)$ in a pertubative scalar field theory, we can take derivatives with respect to $t_i$ and get correlators involving the conjugate field $\pi(t_i, \mathbf{x}_i)$.[10] Thus, from the perspective of worldline gravity, computing correlators of $\pi$ involves computing the change in the path integral when we perturb the boundary conditions $J \in \mathfrak{S}$. Note that the canonical commutation relation between $\phi$ and $\pi$ could be determined from these correlators.

Thus, it is an interesting question to ask in universe field theory whether there are any singularities in the operator product $\phi(J_1) \phi(J_2) = \widehat{Z[J_1]} \widehat{Z[J_2]}$ in the coincident limit $J_1 \to J_2$. If this operator product is singular, generically there should be operators acting on $\mathcal{H}_{\text{BU}}$ that do not commute with $\widehat{Z[J]}$. If the theory is furthermore weakly coupled, we could take derivatives along some direction in $\mathfrak{S}$ and construct an analog of an operator $\pi(J)$, which is canonically conjugate to $\phi(J)$. On the other hand, if operator products in the universe field theory are completely non-singular, the operator algebra will be Abelian. This latter possibility would be a desirable result, because, in gravity theories that have a boundary dual (which in general, could be a disorder-averaged theory), it is hard to give a boundary interpretation to the operators that do not commute with $\widehat{Z[J]}$ [17].

These OPE-type coincidence singularities exist for loop operators [39] in minimal string theory, at least in the genus expansion. Concretely, the operator to consider is the density of eigenvalues of the matrix integral, written in third-quantized notation as $\widehat{\rho(x)}$. The operator $\widehat{\rho(x)}$ is just a specific example of the general $\widehat{Z[J]}$ operator. The eigenvalue direction $x$ of the dual matrix integral is known to be the target space coordinate for $\mathfrak{S}$ in minimal string theories [40], and is thus the appropriate "position-space" variable to diagnose the coincidence limit. Perturbatively, we have $\widehat{\rho(x)}\widehat{\rho(y)} \sim -(x-y)^{-2}$; see, for example, equation (139) in [15].[11] Akin to singularities in a BCFT as operators approach the boundary, there are also singularities in the loop operators as $x \to 0$, which is a boundary of $\mathfrak{S}$ in perturbation theory. Thus, at least perturbatively, there exist operators that do not commute with $\widehat{\rho(x)}$. In the string field theory of the $c = 1$ matrix model [41] this becomes completely explicit: The density of eigenvalues is taken to be the field variable, and this field also has a canonical momentum. It is interesting that, non-perturbatively, the two-point function of $\widehat{\rho(x)}$ gets corrected and is replaced by the sine-kernel which washes out the $(x-y)^{-2}$ singularity. It thus remains an open possibility that non-perturbative effects in gravity remove all such OPE-type singularities, and all the $\widehat{Z[J]}$ operators commute.

---

[10]We thank Douglas Stanford for asking us about the canonical momentum operator.

[11]We thank Douglas Stanford for pointing this out.

## 2.2 Relation to the GNS construction

The procedure described in [17] for constructing the third-quantized Hilbert space is analogous to the Gelfand-Naimark-Segal (GNS) construction in algebraic quantum field theory; see theorem 2.34 of [42] for a brief explanation of the GNS construction and for more references. In the GNS construction, we are given an algebra of operators and a state $\omega$, which is defined abstractly as a positive linear map from the operator algebra into $\mathbb{C}$. In [17], the algebra includes all sums of products of the field operators $\phi(J)$, and the state is the linear map that maps the product $\phi(J_1)\dots\phi(J_n)$ to the complex number computed by the worldvolume path integral with boundaries corresponding to $J_1,\dots,J_n$. The GNS method first constructs a pre-Hilbert space, which is the span of all formal kets $|A\rangle$, where $A$ is any operator in the algebra. The inner product is defined as $\langle A|B\rangle := \omega(A^*B)$. This pre-Hilbert space contains null vectors, which are modded out to give the GNS Hilbert space.[12] This makes the analogy to [17] clear. Notice that this construction of the Hilbert space takes as input *all* correlation functions in a *single* state. See also the recent paper [30] for more on the relationship between the GNS construction and baby universes.

## 3 Global states in the third-quantized Hilbert space

In this section, we discuss various possibilities for the choice of a global state in the third-quantized Hilbert space $\mathcal{H}_{\text{BU}}$.

Hartle and Hawking [25] constructed a particular state $|\text{HH}\rangle$ in the third-quantized Hilbert space, as follows. Consider a $D$-dimensional theory of gravity. Let $Y$ denote a closed $(D-1)$-dimensional manifold, *not necessarily connected*. The Hartle-Hawking state is a function that assigns a complex number to each closed $(D-1)$-manifold $Y$

$$\Psi_{\text{HH}}(Y) := \int_{X:\partial X=Y} e^{-I(X)}, \tag{5}$$

where the integration is to be done over all $D$-manifolds $X$ such that $\partial X = Y$. We work with unnormalized states and density matrices. The analogous object in the worldline theory is

$$\Psi_{\text{KG-vac}}(x_1,\dots,x_n) := \int_{X:|\partial X|=n} e^{-I_{\text{WL}}}, \tag{6}$$

where $X$ is a one-dimensional manifold and $x_i \in \mathbb{R}^{1,d-1}$. Note that $\Psi_{\text{KG-vac}}$ depends on the tuple $(x_1,\dots,x_n)$, and the non-negative integer $n$ is arbitrary. Note that (5) is equal to the correlation function $\langle\text{HH}|\phi(w_1)\dots\phi(w_n)|\text{HH}\rangle$ where $w_1,\dots w_n$ are the connected components of $Y$, and that (6) is equal to the correlation function $\langle\text{KG-vac}|\phi(x_1)\dots\phi(x_n)|\text{KG-vac}\rangle$. Both (5) and (6) can be thought of as expressing the wavefunction in an overcomplete set of non-orthonormal states. The Klein-Gordon vacuum state would usually be written in the orthonormal basis of eigenstates of the field operators $\phi(\mathbf{x})$ as $\langle\phi(\mathbf{x})|\text{KG-vac}\rangle$, but (6) is also correct, and is directly analogous to (5) [25]. See section 2.2 for more details about how all correlation functions in a single state encode the full Hilbert space. The analog of $\langle\phi(\mathbf{x})|\text{KG-vac}\rangle$ in the universe field theory is $\langle\alpha|\text{HH}\rangle$ [17].

In [27], Page proposed a state for the universe, which is *different* than the state proposed by Hartle and Hawking in [25]. Page's state is a density matrix $\rho_{\text{Page}}$ whose components (in an overcomplete non-orthonormal basis) are given by

$$\rho_{\text{Page}}(Y_1,Y_2) := \int_{X:\partial X=\overline{Y_1}\cup Y_2} e^{-I(X)}. \tag{7}$$

---

[12]Technically, one also needs to add appropriate limit points to ensure completeness (in the sense of a metric space).

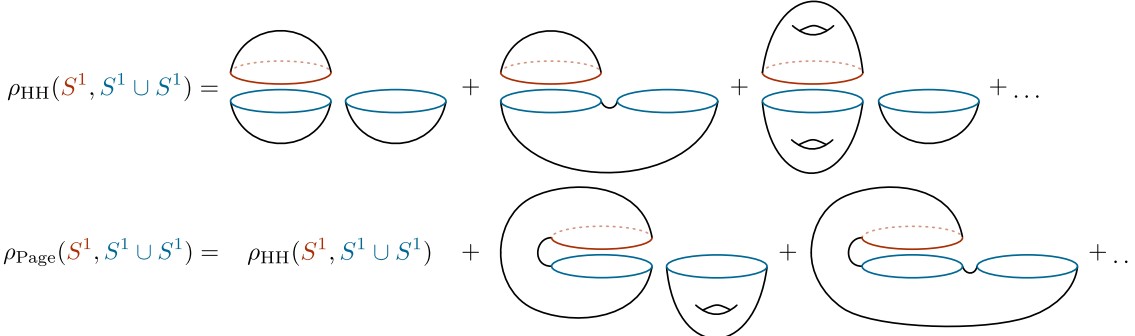

$$\rho_{\text{HH}}(S^1, S^1 \cup S^1) = \quad + \quad + \quad + \dots$$

$$\rho_{\text{Page}}(S^1, S^1 \cup S^1) = \quad \rho_{\text{HH}}(S^1, S^1 \cup S^1) \quad + \quad + \quad + \dots$$

Figure 1: The difference between the Page density matrix $\rho_{\text{Page}}$ and the density matrix $\rho_{\text{HH}}$ corresponding to $|\text{HH}\rangle$. This figure depicts contributions to a particular matrix element of these density matrices in two-dimensional gravity where a general $Y$ is a union of disconnected circles. In red we indicated the bra and in blue the ket. In the Page density matrix (7) [27], bra-ket wormholes are present, whereas they are absent in the density matrix for the Hartle-Hawking state (10) [25]. In particular, the set of configurations that contribute to $\rho_{\text{Page}}(Y_1, Y_2)$ is a superset of the configurations that contribute to $\rho_{\text{HH}}(Y_1, Y_2)$.

Again, we emphasize that neither $Y_1$ nor $Y_2$ is required to be connected, and in general they have different numbers of connected components. The overline indicates orientation reversal and accounts for complex conjugation in the bra. Note that the right hand side of (7) is the same as the object $\langle Y_1 | Y_2 \rangle$. The punchline of [27] is that because there exist Euclidean wormholes $X$ that connect $Y_1$ and $Y_2$, these configurations enter the path integral on the right hand side of (7) and render the state $\rho_{\text{Page}}$ mixed.

Next, we actually note that[13]

$$\rho_{\text{Page}} = \mathbb{1}_{\mathcal{H}_{\text{BU}}}, \tag{8}$$

where $\mathbb{1}_{\mathcal{H}_{\text{BU}}}$ is the identity matrix on the baby universe Hilbert space.[14] This is true, because, according to (7), $\rho_{\text{Page}}(Y_1, Y_2) = \langle Y_1 | Y_2 \rangle$ with *nothing* else inserted in the path integral. In particular, if $\dim \mathcal{H}_{\text{BU}} > 1$, we see that

$$\rho_{\text{Page}} \neq |\text{HH}\rangle\langle\text{HH}|. \tag{9}$$

In more detail, note that the components of the density matrix corresponding to $|\text{HH}\rangle$ are given by

$$\rho_{\text{HH}}(Y_1, Y_2) = \Psi_{\text{HH}}(Y_1)^* \, \Psi_{\text{HH}}(Y_2). \tag{10}$$

The right hand side of (10) is a product of two independent path integrals, *without any wormholes connecting $Y_1$ and $Y_2$*. This is simply because $\Psi_{\text{HH}}(Y_1)$ and $\Psi_{\text{HH}}(Y_2)$ are well-defined objects that have already been defined in (5). We cannot add additional wormholes between $Y_1$ and $Y_2$. The difference between $\rho_{\text{page}}$ and $\rho_{\text{HH}}$ is illustrated in figure 1, as also originally noted in [27].

We believe this point should be non-controversial, but nevertheless it is important to highlight because it has important consequences. For instance, there has been much recent work on the the spectral form factor in JT gravity, and its non-factorization $\langle Z(\beta_1)Z(\beta_2)\rangle \neq \langle Z(\beta_1)\rangle\langle Z(\beta_2)\rangle$ [13–15]. Here the left hand side is equal to

---

[13]We thank Don Marolf and Henry Maxfield for discussions on this point.

[14]Note that this density matrix is ill-defined if the Hilbert space is infinite dimensional.

$\langle HH|\widehat{Z(\beta_1)}\widehat{Z(\beta_2)}|HH\rangle$, whereas the right hand side is equal to $\langle Z(\beta_1)|HH\rangle\langle HH|Z(\beta_2)\rangle$. If we allowed wormholes connecting $Y_1$ and $Y_2$ on the right hand side of (10), we would also say that $\langle Z(\beta_1)|HH\rangle\langle HH|Z(\beta_2)\rangle$ is computed by a two-boundary quantity with all possible wormholes connecting the two $Z$ insertions. This sum would thus be exactly the same as $\langle HH|\widehat{Z(\beta_1)}\widehat{Z(\beta_2)}|HH\rangle$, and there would be no factorization puzzle.

If we allowed wormholes between products like that on the right hand side of (10), the variance of all boundary-creating operators (which are all the operators) would be zero, and we would conclude that the third-quantized Hilbert space is one-dimensional. Even though it would be desirable to prove that $\dim \mathcal{H}_{BU} = 1$,[15] this line of argumentation of connecting everything to everything is flawed.

Let us also comment on the density matrix considered in Hawking's paper [44], which was contemporaneous to [27]. Hawking considers tracing out all connected components of the spatial manifold except "our own" connected universe, and interprets this object as the density matrix of the universe in which we live. This is necessarily an approximate notion of a density matrix that would only make sense, for instance, at late times in de Sitter space. See also section 4.3 below. Instead, the object considered by Hawking is more appropriately identified as a two-boundary correlation function in the state $|HH\rangle$.

# 4 Implications

In the previous section we saw that the choice of a global state in the third-quantized Hilbert space defines what bra-ket wormholes can contribute. We will now discuss two applications. The first is to clarify what wormhole contributions are present in entropy computations, and the second is to discuss how approximate wormholes could arise when restricting to a single large connected universe.

## 4.1 Bra-ket wormholes in entropy computations

Let us say we want to compute the quantity $(\mathrm{Tr}\rho)^n$ where $\rho$ is a state on $\mathcal{H}_{BU}$. The first point we want to emphasize is that before we try to compute $(\mathrm{Tr}\rho)^n$, we should first specify which $\rho$ we are talking about. In general, $\dim \mathcal{H}_{BU} > 1$ and the global state of the universe can be chosen from an infinite number of possibilities. So, let us pick a particular $\rho$, which means that we have a definite rule for computing $\rho(Y_1, Y_2)$ for each choice of $Y_1$ and $Y_2$. For example, we could pick $\rho_{Page}$ which is defined in (7), or we could pick $\rho_{HH}$ which is defined in (5) and (10).

Now, following the exact same logic as in the previous section, while computing $(\sum_Y \rho(Y, Y))^n$ we may not freely add wormholes connecting an arbitrary subset of the various connected components of the $2n$ insertions of $Y$. By specifying the state $\rho$ in question, the quantity $\rho(Y, Y)$ is completely well-defined (and it may or may not include bra-ket wormholes depending on what $\rho$ we picked). In particular, the definition of $(\sum_Y \rho(Y, Y))^n$ is unambiguous: We should simply compute $\rho(Y, Y)$ for each $Y$ using the particular gravitational path integral in the definition of $\rho$, do the explicit sum over $Y$ by hand, and raise it to the $n$-th power.

Similar comments apply to $\mathrm{Tr}(\rho^n)$. The quantity $\sum_{Y_1,...,Y_n} \rho(Y_1, Y_2)...\rho(Y_n, Y_1)$ is unambiguously defined once we have specified the gravitational path integral that computes $\rho(Y_1, Y_2)$.

It might seem like $\mathrm{Tr}(\rho^n)$ and $(\mathrm{Tr}\rho)^n$ are both computed by the same gravity path integral

---

[15]This statement has recently been included as a swampland conjecture [43].

[12],[16] leading one to believe that they are equal for any choice of $\rho$. However, this reasoning is incorrect, because the only wormholes that can possibly appear are in the computation of the individual quantities $\rho(Y_1, Y_2)$, and then we should use the sums $\sum_{Y_1,...,Y_n} \rho(Y_1, Y_2)...\rho(Y_n, Y_1)$ and $(\sum_Y \rho(Y, Y))^n$ as the definitions $\text{Tr}(\rho^n)$ and $(\text{Tr}\rho)^n$, respectively. For example, to reiterate our basic point, bra-ket wormholes are present in the gravitational path integral that computes $\rho_{\text{Page}}(Y_1, Y_2)$, while they are not present in the gravitational path integral that computes $\rho_{\text{HH}}(Y_1, Y_2)$.

Similarly, before discussing wormholes of the sort in section 6 of [12] or in [29], which refer to one-universe density matrices, we first need to specify the global state $\rho$ in the universe field theory. In section 4.3, we discuss how such wormholes can be "emergent" in special cases, even if they are not present in the original definition of $\rho$.

## 4.2 Relation to entropy computations in holographic QFTs

The Ryu-Takayanagi formula [45–49] and its recent extensions involving "bath" regions with non-dynamical gravity [11,12,50–52] compute entropies defined purely in the dual quantum-mechanical system which has no gravity. Conceptually, this is a very different quantity than computing the entropy of $\rho_{\text{Page}}$ or $\rho_{\text{HH}}$. The density matrix whose entropy we want to compute in the AdS context is a state in the *boundary* Hilbert space, whereas $\rho_{\text{HH}}$ and $\rho_{\text{Page}}$ are states in the baby universe Hilbert space $\mathcal{H}_{\text{BU}}$. This is a fundamental difference, and we will denote density matrices in the boundary (plus bath, if present) Hilbert space by $\rho_{\text{CFT}}$.

For entropy calculations in a holographic QFT (plus bath, if present), we follow [47] to first set up the replica trick purely in the non-gravitational description, and then ask what is the gravity dual of this calculation. In particular, in the third-quantized language of [17], we would be computing expectation values of the Renyi-entropy "operator", i.e. quantities like $\langle \text{HH}| \widehat{\text{Tr}(\rho_{\text{CFT}}^n)} |\text{HH}\rangle$. The Renyi entropies of boundary density matrices, or individual matrix elements of $\rho_{\text{CFT}}$, become third-quantized operators acting on $\mathcal{H}_{\text{BU}}$. This is exactly analogous to the operator on the left hand side of equation (3.42) in [17]. By definition of $|\text{HH}\rangle$, there are wormholes connecting all the boundaries in the computation of $\langle \text{HH}| \widehat{\text{Tr}(\rho_{\text{CFT}}^n)} |\text{HH}\rangle$.

In the context of this paper or section 6 of [12], the quantities under consideration were the entropies of states in $\mathcal{H}_{\text{BU}}$, i.e. quantities of the form $\text{Tr}(\rho_{\text{HH}}^n)$ or $\text{Tr}(\rho_{\text{Page}}^n)$. In AdS, the quantities analogous to this would be

$$\sum_{\{\beta_1...\beta_n\}} \langle Z(\beta_1)Z(\beta_2)\rangle \langle Z(\beta_2)Z(\beta_3)\rangle ... \langle Z(\beta_n)Z(\beta_1)\rangle. \tag{11}$$

In this quantity there are no wormholes connecting all $2n$ of these insertions, but only wormholes connecting the two $Z$'s within a single $\langle \cdots \rangle$.

## 4.3 Comments on single-universe observables

In inflationary physics, the two-point function of perturbative fields in a single connected universe plays a central role [53]. There is a regime in which such observables should be well-defined: a large classical universe with highly suppressed topology fluctuations, in which we can make sense of inflationary correlators, as well as the density matrix discussed in [12,44]. In the worldline gravity example considered in section 2, and allowing worldlines allowed to split, this approximation is valid when the particles are very heavy and particle production is suppressed. In this regime, the position and momentum operators of a single particle become well-defined.

---

[16]Note that [12] restricted to the case when $Y$ is connected, see section 4.3 for this case.

$$\sum_Y \rho_{\text{HH}}(w \cup Y, w' \cup Y) =$$

Figure 2: Representation of $\rho_{\text{1-univ}}(w, w')$ in (12) with $\rho = \rho_{\text{HH}}$. Recall that $Y$ is not necessarily connected. For an appropriate choice of state and model, the sum over $Y$ can give rise to an approximate "effective" wormhole between $w$ and $w'$ [28, 54].

A single connected (late-time) universe is the scenario considered in section 6 of [12] and in [29], so let us see what our observations imply about this setting. The idea is that we want to define a density matrix for a single universe, for instance via

$$\rho_{\text{1-univ}}(w, w') := \sum_Y \rho(w \cup Y, w' \cup Y), \tag{12}$$

where $w$ and $w'$ label a complete set of configurations of single-universe states, and $Y$ denotes the configurations of the "other" universes.[17] Note that this split between our universe and other universes becomes arbitrarily good in an appropriate regime, such as late times in de Sitter. Note that there is *still* some choice of $\rho$ involved on the right hand side of (12), which could be $\rho_{\text{HH}}$, $\rho_{\text{Page}}$, etc.

We can now ask the question whether the sum over $Y$ in (12) leads to an effective wormhole between $w$ and $w'$, even for states like $\rho_{\text{HH}}$ as in (10) which do not have an explicit wormhole between $w$ and $w'$ in their fundamental definition. As pointed out in [28], having effective wormholes is possible, though this likely depends on the choice of the state $\rho$ and the details of the gravitational model.

The bra-ket wormhole discussed in equation (6.2) of [12] plays an important role in avoiding a bags-of-gold type paradox in de Sitter space. Such wormholes have also recently been shown to prevent violations of strong subadditivity of entropy [29]. Our view is that these wormholes, if not already present in $\rho$, must be emergent: The sum over $Y$ in the definition (12) of $\rho_{\text{1-univ}}(w, w')$ should lead to approximate geometric connections between the bra $w$ and the ket $w'$, as in figure 2. In principle, this could happen even for states such as $\rho_{\text{HH}}$ that do not have these bra-ket type wormholes in their definition. Note that this approximate geometric connection will generically not be equal to the original cylindrical geometry between $w$ and $w'$, if the latter exists in the definition of $\rho$ (an example of such a case would be $\rho_{\text{Page}}$). This is analogous to the diagonal piece in the sum over periodic orbits in the spectral form factor giving rise to a cylinder [12, 15], and the "diagonal = cylinder" identity derived in the setup of [28]. See also [54].

A more drastic possibility is for $\rho_{\text{HH}}$ to be non-perturbatively gauge equivalent to some state with bra-ket wormholes (which need not be $\rho_{\text{Page}}$).[18] An extreme possibility is that

---

[17]The equation (12) is correct even though the $Y$'s do not form an orthonormal basis. This is related to the fact, nicely explained in [17], that the path integral computes inner products, and sidesteps the complications of specifying an orthonormal basis for Hilbert space. In other words, cutting the gravitational path integral still provides us with a resolution of the identity, albeit in a non-orthonormal basis.

[18]This might seem surprising, but the gauge redundancies in gravity are strong [17, 55] and need to be explored

dim $\mathcal{H}_{\text{BU}} = 1$, a condition which is equivalent to the condition that $\rho_{\text{HH}}$ and $\rho_{\text{Page}}$ be gauge equivalent to each other.[19]

In order to compute the Renyi entropies of $\rho_{\text{1-univ}}$, one would follow the idea described in section 4.1. One would first compute the matrix elements $\rho_{\text{1-univ}}(w, w')$ using (12), and then explicitly compute the sums in the index contractions in $\text{Tr}(\rho_{\text{1-univ}}^n)$.

The recent paper by Giddings and Turiaci [56] also contained some expressions similar to ours. Their equation (3.5) is similar to our (12), and their equation (3.8) is arguing for computing the Renyi entropies of $\rho_{\text{1-univ}}$ just like we have described. However, we believe that the LHS of their equation (3.10) should be replaced by $\langle \text{HH} | \widehat{\text{Tr}(\rho_{\text{CFT}}^n)} | \text{HH} \rangle$, like we discussed in section 4.2. In particular, we are in complete agreement with [11,12] for computing Renyi entropies of $\rho_{\text{CFT}}$ (states of the boundary system, possibly coupled to a bath).

A brief comment about entanglement entropy in the worldvolume theory. Usually, one computes the entanglement properties of a QFT by considering a complete set of states on a Cauchy slice in spacetime. But one could imagine special states in which the entanglement could be computed by considering worldlines of heavy particles. For instance, the particles could have two-internal states, and we might consider an EPR state of two such heavy particles. See [57] for some work in this direction. The same comments should apply to the universe field theory; there should be a limit in which the entropies of subregions on the worldvolume become well-defined.

# Acknowledgments

We thank Dionysios Anninos, Austin Joyce, Adam Levine, Juan Maldacena, Don Marolf, Henry Maxfield, Edgar Shaghoulian, Steve Shenker, Douglas Stanford, John Stout, Zhenbin Yang and Ying Zhao for insightful discussions, and Edgar Shaghoulian for comments on the draft. TA is supported by the Delta ITP consortium, a program of the Netherlands Organisation for Scientific Research (NWO) that is funded by the Dutch Ministry of Education, Culture and Science (OCW). JK is supported by the Simons Foundation. This research was supported in part by the National Science Foundation under Grant No. NSF PHY-1748958.

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
