# Peer review of "Density matrices in quantum gravity"

_SciPost Physics, doi:SciPost Phys. 9, 045 (2020)_

## Round 1 · Referee Report · Anonymous (Referee 1) · 2020-9-9

Report

The paper makes several important points regarding recent work on spacetime wormholes and baby universes, connecting previous work from various authors and perspectives. The paper is well organized and written, and the arguments are presented in a clear way. I recommend publication.

---

## Round 1 · Referee Report · Anonymous (Referee 2) · 2020-9-12

Strengths

1. Builds on the earlier work in [17] to help clarify various issued in "third-quantisation" (or, to use the better name introduced in the paper, "universe field theory")

2. Excellent use of the familiar example of Klein-Gordon theory/worldline gravity to understand general phenomena that can show up in this framework.

Weaknesses

1. A couple of minor conceptual distinctions could be emphasized more (see below)

Report

This is a strong paper that builds on recent progress in providing much needed clarity to questions about the quantum mechanics of universe creation and annihilation. The authors start with the example of Klein-Gordon theory/ worldline gravity and then use intuition from this example to conjecture how things should work in higher dimensions. Finally they discuss the distinction between the Page density matrix and the (pure) Hartle-Hawking state. Subject to the minor revisions listed below, I believe that it should be accepted for publication.

Requested changes

1. A central claim of the paper is that the Hartle-Hawking state does not contain bra-ket wormholes. By this, the authors mean that there are no wormholes connecting the bra <HH| (in the baby universe Hilbert space) and the ket | HH> (in the baby universe Hilbert space) if both appear in the calculation. However, it doesn't mean there aren't wormholes connecting a "bra" boundary and a "ket" boundary when evaluating a boundary partition function (e.g. Tr(rho^n) as the expectation of a baby universe Hilbert space operator in the Hartle-Hawking state. This is discussed in section 4.2, but it would be good if it was emphasized much earlier. For example, my interpretation of Section 6 of [12] is that it is discussing the latter, not the former, although that the wording is definitely ambiguous (probably because [12] was published before [17]).

2. Similarly, the paper makes the important point that the commutativity of boundary creation operators is slightly more subtle than was suggested in [17], using the example of time-like separated Klein-Gordon field operators. However there are really two distinct notions of commutativity here. The first is the order in which boundaries are added. The second is the existence of singularities and branch cuts in the correlation functions when two operators are inserted with no Euclidean time separation. It is only in the second sense (which corresponds to operators not commuting in the canonical quantisation of Klein-Gordon theory) that boundary creation operators don't commute. No matter which order we apply boundary creation operators, we always get the Euclidean time ordered correlation functions. So the boundary creation operators commute in the first sense. Again this is clear from a thorough reading of the paper (and is clearly understood by the authors), but could be emphasized more clearly at the beginning.

3. The density matrix rho_Page needs to be proportional to the identity matrix. However, in many/most relevant cases, the baby universe Hilbert space is infinite-dimensional, so no such density matrix exists. Some brief comments on this would be nice.

---

## Round 2 · Author Response

We would like to thank both reviewers for their encouraging and detailed comments. We hope to have addressed the reviewer's with the changes listed below.

---

## Round 2 · List of Changes

1. To give an early disclaimer that expectation values in the Hartle-Hawking state do not have wormholes connecting the sandwiched |HH>'s, but that other wormholes may exist when acting on the third quantized Hilbert space, we have added the following sentence at the end of the introduction:

"In section 4.2 we discuss how boundary entropy computations in AdS are conceptually different, and emphasize that the replica wormholes in that case are simply the wormholes between the various ket boundaries in the bulk Hartle-Hawking state."

  1. To clarify the points about commutativity we have added to the introduction:

"For example, the Klein-Gordon field operators do not commute at timelike separation. However, as usual, if the target space (which, in general, would be “superspace”) is analytically continued to Euclidean signature, we do expect the boundary-creating operators to commute."

as well as footnote 4:

"Superspace has infinitely many negative-signature directions and this analytic continuation might be subtle."

  1. We have added footnote 14:

"Note that this density matrix is ill-defined if the Hilbert space is infinite dimensional"

when discussing the page density matrix being proportional to the identity matrix. This is done in order to address the referee's concern that such a density matrix does not exist in most relevant cases.

---

## Editorial Decision

published